# The Response of Social Crime Prevention Police to Cyberbullying Perpetrated by Youth in Rural Areas of South Africa

**DOI:** 10.3390/ijerph182413421

**Published:** 2021-12-20

**Authors:** Fani Radebe, Michael Kyobe

**Affiliations:** 1Department of Computer Science and Informatics, University of Free State, Bloemfontein 9300, South Africa; 2Department of Information Systems, University of Cape Town, Cape Town 7700, South Africa; michael.kyobe@uct.ac.za

**Keywords:** design science, adolescents, development, cyberbullying, cross-age cyberbullying, mobile response system, peer nominations

## Abstract

Recently, South Africa has seen a surge in violence, cyberbullying by learners against peers, and online malicious acts against teachers. In response, the South African Department of Basic Education invited the social crime prevention police to intervene. This study reports on the developmental issues contributing to cyberbullying and the police response to this violence in rural schools. An extensive literature review was conducted, and a conceptual framework was developed to guide the study and development of a mobile application. This framework was tested using data collected from focus groups, 8 police officers, 9 teachers, 52 grade-10 learners, and 27 grade-12 learners. The data were analyzed using thematic and quantitative techniques. The findings reveal some developmental issues. For instance, teachers are often targeted by learners online because they fail to take prompt action when learners report cyberbullying incidents. This finding is consistent with the developmental theory which predicts that lack of support would create a permissive context for cyberbullying. In addition, the popularity of cyberbullying has a stronger influence on older, rather than younger, adolescents. Older adolescents are more concerned about gaining popularity than being socially accepted. Recommendations are made which can be useful to schools, learners, and the police force in their fight against cyberbullying.

## 1. Introduction

South African public schools have seen a rise in violence against teachers, including cyberbullying. Around the world, violence and bullying are part of reality for teachers and learners alike [1,2]. School violence often arises from unresolved physical and cyberbullying incidents [3,4,5]. In response to the school violence, the Department of Basic Education invited the South African Police to help restore safety in schools. Each school has a designated police officer to help address social crime-related incidents that arise in school premises. Generally, one police officer is assigned to one or a cluster of schools. Although the involvement of the police in addressing school violence is undoubtedly essential, their role against cyberbullying remains unclear [6]. The factors that exacerbate cyberbullying challenges include anonymity and distancing. Cyberbullies do not always witness the harm caused by their actions (distancing) due to online attack characteristics, including lack of face-to-face contact and creation of empathy for the victim [1]. The negative outcomes of cyberbullying can be both physical and psychological, including increased stress and alcohol abuse [7]. In South Africa and abroad, teachers have experienced bullying by their pupils, which is detrimental to a safe learning environment, including low morale and motivation, and negative emotions [1,8,9]. Up to 6.6% of teachers in 153 Tshwane secondary schools experienced cyberbullying by their learners in the form of rumour spreading and gossip [9]. The use of mobile devices for Internet access is much more common in South African rural areas compared to other methods [10], which translates to engaging in cyberbullying [11]. While research on mobile bullying is lacking, the high mobile phone usage in rural areas calls into question the applicability of existing cyberbullying theories in rural settings [12]. Notably, cyberbullying modes differ across age demographics. Cyberbullying often takes place in online gaming among pre-teens, whereas adolescents experience it on social networking sites, with Facebook being the most popular platform [13]. Cyberbullying normally involves ganging up with intentions to harm others psychologically, such as causing embarrassment, spreading lies, and reputational damage online through impersonation [14]. The prevalence of cyberbullying among young adults (15–18 years) was found to be higher than older adults who are 26–35 years [7].

Generally, most cyberbullying elements are criminal in nature, including threats of violence, criminal intimidation, stalking, hate crimes, and sexual harassment, all of which could be prosecuted if brought to court. Schools, however, have a greater institutional responsibility, rather than only focusing on the criminal liability of learners [15]. In South Africa, adults rely on human rights and the Protection from Harassment Act 17 of 2011, but its suitability in addressing cyberbullying is questionable [2]. Only the Children’s Act makes explicit reference to bullying but lacks definitions for bullying and cyberbullying. Although bullying is not yet criminalised in South Africa [2], a new Act, the Cybercrimes Act 19 of 2021 could help address cyberbullying [16]. However, providing evidence that links perpetrators to cyberbullying incidents is still a challenge due to the anonymity affordance of social network technologies, which makes it hard to catch and bring cyberbullies to face the consequences of their actions [13,17]. Additionally, cyberbullying may appear as an unintentional act, such as forwarding of photos that were initially provided without any intention to bully from the source but are still harmful to victims [18]. Therefore, identifying perpetrators of cybercrimes and bullying is challenging, and these observations signal the need for an intervention that provides a platform to safely report cyberbullying in schools.

Reporting is essential in countries such as South Africa, where the crime rate is one of the highest in the world [12,19]. However, reporting is still a challenge, making it difficult to address traditional bullying and cyberbullying in secondary schools. The lack of reporting stems from fear of further victimisation or retribution and overreaction by adults, and lack of awareness, while teachers try to protect their profession and avoid the stigma associated with finding unruly learners in their care [2,8,13,20,21]. On the other hand, if unchallenged due to a lack of reporting, bullies may assume impunity in carrying on their behaviour, while victims continue to suffer low self-esteem and suicide ideation, and underperform academically [8,22]. As a result, teachers may quit their profession prematurely [8].

Early diagnosis is paramount in efforts to curb cyberbullying behaviour in schools, [21,23,24]. Mechanisms such as self-report and peer nomination are used to identify victims of bullying [25]. These mechanisms may, however, lead to incongruent findings about the prevalence of cyberbullying, due to the bias and subjectivity of self-reporting, and the lack of subjective experience in peer nominations [26]. Therefore, there is a need for adequate and direct cyberbully identification mechanisms.

The knowledge of cyberbullying is still low and therefore interventions are still being developed and differ from those for traditional bullying [1,11]. The prevalence of traditional bullying and violence against teachers has received much attention; however, cyberbullying remains unclear. For instance, gender differences across life span and the occurrence of cross-age (young to older) cyberbullying are less clear [7]. There is no agreed-upon definition of learners-to-teachers (cross-age) bullying, but its nature includes undermining, disempowerment, and negative effects on mental health of teachers [1,9]. The current study adopts the description of learners-to-teachers online violence as online “malicious acts” against teachers including the characteristics mentioned above [27] (p. 195). Additionally, “more research is needed that makes direct age comparisons regarding cyberbullying and victimization”, and understanding predictors of cyberbullying behaviour among adolescents may help reduce its occurrences in schools [13] (p. 29) [28]. The understanding of the police’s role in reducing cyberbullying is “especially important in schools, generally seen as the preparation for adult life” [22] (p. 2). Therefore, this paper investigated the issue from developmental perspective predictors of cyberbullying between learners and online malicious acts between learners and their teachers (cross-age) in rural schools in South Africa.

## 2. Literature Review

### 2.1. Cyberbullying

The definition of cyberbullying relates to that of traditional bullying as a harmful behaviour repeated over time with a clear power imbalance between the bully and victim [13]. The rapid and wide spread of harmful content, as well as bullies’ ability to conceal their identity (anonymity) through online technologies, and relates to repetition and power imbalance. The use of various social networking sites and mobile devices makes it hard to define cyberbullying [1,23]. However, cyberbullying has distinctive characteristics that allow invasion of victims’ privacy at any time. The use of mobile devices to threaten others through messages, phone calls, and sending obscene images is referred to as mobile bullying, which is a subset of cyberbullying [19].

### 2.2. Cyberbullying Roles

Cyberbullying roles can be categorised into pure bullies, bully-victims, and victims. Pure bullies exhibit aggressive behaviour and seldom fall victim to bullying, whereas bully-victims are learners who are reactive in nature, who bully when bullied by others [3,23]. Victims are learners who cannot easily defend themselves against bullies [13]. The identification of individual learners’ roles in cyberbullying is essential to inform targeted interventions, instead of using broad-brush approaches with no specific cyberbullying elements [3,22].

### 2.3. Cyberbullies’ Sociometry

Social scientists examine social networks to understand macro-level patterns of people’s interactions in groups, which are shaped by their actions in a way that produces certain outcomes [29]. Peer acceptance is inferred using sociometric statuses including popularity, power and dominance, rejection, being neglected, and controversial [30,31,32], and a sociogram is used to visualise the outcomes. Status also refers to power and dominance by gaining respect from peers [33]. Sociograms are graphs that consist of nodes and links (in- and out-degrees) between node ties. Nodes can represent individuals in a group such as learners, and their interactions or ties are represented with links [34]. Similarly, cyberbullies and their roles can be identified using peer nominations and constructing a sociogram to reveal their behaviour through peers’ perceptions (peer nominations). Nominations are interpreted to identify cyberbullies and their specific roles [34]. The definition of cyberbullying is given to aid nominations of learners who fit the description of cyberbullying, and questions that seek to reveal the form of bullying victimisation, and the identification of perpetrators [34]. The number of nominations received indicates learners’ popularity regarding cyberbullying behaviour [34,35]. Therefore, “bullying can be associated with popularity” and interventions that include the identification of popular cyberbullies could influence behavioural change [33] (p. 150) [36]. Similar to the use of the number of received nominations, popularity can also be determined using PageRank. PageRank is a Google algorithm that is used to measure websites’ popularity or influence based on a node’s connection to other well-connected nodes [37]. That is, a node’s connection to a number of well-connected nodes contributes a higher score than a connection to the same number of nodes that are not well connected. To measure cyberbullying popularity using peer nominations in this study, the researchers chose the PageRank sociometric measure because it is based on a voting concept [38].

Social choices can be made subjectively or objectively. Subjective choices are guided by intuitive feelings, such as liking or disliking others on first impression. Objective choices are based on experience and knowledge, such as knowing that a person has or does not have skills for a particular group task [39]. Hence, providing a specific criterion enables participants to make informed nominations [39]. Since bullying is an objective and conscious action, and both bullies and victims are disliked by peers [32], the researchers chose objective criteria for the identification of cyberbullies in classrooms, where learners can be asked to nominate their bullies or victims. The uniqueness of peer nominations is that it reveals incidents that are not easily observed by teachers and parents [40]. The nomination result can be used in social network analysis (SNA) to provide a big-picture view with the ability to examine and understand interactions between actors within the context of operation [41].

### 2.4. Social Network Analysis (SNA)

Mechanisms that are used to identify cyberbullies for intervention include self- and peer-nominations [34]. Social network analysis (SNA) can be applied to developmental questions, especially those stemming from developmental system theoretical views [41]. Therefore, a positive development of adolescents can be controlled and evaluated through social network analysis to inform suitable intervention programmes. The authors of [41] note that SNA basic principles focus on (1) a group of actors and their relationship; (2) evaluation of characteristics of actors that are informed by relational processes; (3) evaluations that focus primarily on relations between actors; and (4) understanding elements of a social context and influence of the elements on observed characteristics. Therefore, the use and interpretation of peer nominations makes SNA suitable for the identification of cyberbullies and their roles during a specific developmental stage of learners in schools [34]. Peer nominations enable behavioural observations from multiple participants’ views in a group and have a higher ecological validity than self-reports [33].

### 2.5. Effects Assessment

The identification of cyberbullies can be coupled with assessments to learn about the severity (effects), power imbalance, and the nature of bullying, including whether the act was premeditated, goal-directed, reactive, or proactive [34]. Traditional bullying and cyberbullying have similar consequences [13]. Therefore, their impact can be measured in a similar way, and [42] provides a strategy to measure the impact of bullying in three categories: frequency, impact, and obscenity, which require a formal response. Participants are requested to answer assessment questions by selecting verbal scales that have corresponding numerical values of moderate (1); major (2); and severe (3), for each category. The summation of the selected scales determines the level of impact as follows: sums of 3 to 5 indicate a moderate impact; sums of 6 or 7 indicate a major impact; and sums of 8 or 9 indicate a severe impact. A severe impact is also determined if severe (3) is selected in any of the three scales, even if the sum is below 8. These results can be used to inform the level of intervention required, such as involving social workers for severe effects and seeking peer support for moderate effects.

### 2.6. Technological Intervention

Technological interventions need to be developed to aid detection of cyberbullying in schools. Furthermore, Ref. [43] suggests that learners’ behaviour improves when they know that their online activities are being monitored, which helps to eliminate immoral disengagement. Interventions could be role-based for bullies, bully-victims, victims, friends, and teachers. In the bullying process, bullies can be warned about consequences of their actions, victims are encouraged to seek emotional support and are deterred from retaliation, and friends are discouraged from joining in, but to support victims. The editor of [44] notes that research on cyberbullying prevention is relatively new and ranges from universal programmes with limited or no specific elements targeting cyberbullying, to whole school approaches and Internet safety lessons that include cyberbullying. Seemingly, the role of law enforcement in the fight against cyberbullying has not been examined. Since law enforcement’s role is to prevent crime, gaining an insight into their involvement in curbing cyberbullying behaviour would be valuable.

The following sections examine theoretical works that explain the cyberbullying influencing factors and their interactions. Considered in the present study are theories that provide knowledge about the cyberbullying phenomenon, which is the motivation for constructing the artefacts to solve existing problems [45]. The variables of the theory of planned behaviour (TPB), especially attitude and perceived behaviour control, do not always have consistent results; the TPB is criticised for its inability to fully capture cyberbullying interventions [14]. Therefore, socio-ecological system theory and developmental systems theory (DST) were included in the current study to supplement the TPB elements’ shortfalls in the cyberbullying intervention proposal in the current study.

### 2.7. Theory of Planned Behaviour (TPB)

The theory of planned behaviour (TPB) provides a well-accepted social psychological theory for explaining human behaviour [46]. The TPB suggests individuals’ behavioural decisions are informed by the reasoning process, including attitude, social norms (SN), perceived behaviour control (PCB), and efficacy beliefs. Perceptions of cyberbullying victimisation and perpetration are related to school climate and safety [13]. Therefore, the alignment of school culture to its climate is important to bring about behavioural change to teachers, learners, and other officials [36]. The school culture relates to “assumptions, values, and beliefs”, and climate relates to “actual behavioral change” [36] (p. 159), since the definition of cyberbullying includes intentions to cause harm [13]. The prediction of intentions regarding cyberbullying are mainly facilitated by SN, attitude, and PBC of the theory of planned behaviour [11,14]. SN refers to individuals’ perception of other important people’s expectations. School classrooms are characterised by climate and social norms that may convey the approval of negative conduct among the group [47]. PCB refers to the perception of how easy or difficult the contemplated act is. Attitude relates to the evaluation of an individual’s opinions on the merits and demerits of a contemplated act. Efficacy beliefs relate to the trust that reporting incidents will be successful and without negative consequences [36].

Clearly, increased internet access and experiences of bullying on social media for adolescents suggest a greater likelihood of cyberbullying involvement [11,13]. Hence, if school systems show indifference towards cyberbullying behaviour, perpetrators may assume impunity and continue their behaviour [22]. Attitudinal change towards responses to cyberbullying is important; however, studies on attitudes towards cyberbullying are lacking [11]. SN drives cyberbullying intentions, and together with attitude, they are the most influential elements of preventive intervention designs [14,48]. The TPB is suitable at the developmental period of adolescents, since peer influence contributes significantly to them as a strong SN towards their intention to engage in cyberbullying [48]. Furthermore, at adolescent stages, learners have a permissive attitude towards bullying [32].

### 2.8. Social-Ecological System (SES) Theory

The understanding of the development of cyberbullying in a school context could help the formulation of suitable interventions in schools by noting bullying as an evolving and recurrent behaviour that responds to ecologies and social influences [9,47]. The social-ecological system (SES) theory explains how learners’ personal traits interact with a system or an environmental context to promote or prevent bullying [49]. The SES theory is suited to aiding understanding of nascent bullying behaviour in school settings [47,50]. Cyberbullying behaviour is influenced by complex interactions of factors with individuals’ socio-ecological systems. The significant factors of high rates of cyberbullying perpetration include a lack of clear rules and little teacher support [13], from which young adolescents may justify cyberbullying at class levels [47]. The author of [51] notes that the availability of cyberbullying law (policies) influenced victims’ likelihood of reporting cyberbullying incidents in schools. Contexts such as school administration and institutional infrastructure influence the likelihood of learners’ involvement in cyberbullying [49,50]. Therefore, a positive change to an environmental context could also encourage adolescents to refrain from cyberbullying. This is important because adolescents are more attracted to bullying to gain popularity, than socially accepted behaviour [32].

### 2.9. Developmental Systems Theory

To determine an entry point in terms of age for cyberbullying intervention among learners, developmental factors that characterise cyberbullies were considered. Development is seen as the result of bidirectional person–context interactions. Traditionally, a developmental process is explained in theoretical psychology and theoretical biology “as the result of self-organising processes with emergent properties that have complex, dynamic interactions with environmental influences”, generally denoted as DST [52] (p. 3). These theories posit the temporality and the plasticity of the development system [41]. Plasticity refers to a person’s potential for change, and person–context interactions regulate and constrain plasticity variably over developmental time [41]. For instance, [47] notes that age and cyberbullying justification in a classroom (normative context or shared belief) are developmental factors that significantly influence each other, such that a high justification of cyberbullying between adolescents corresponds to high cyberbullying perpetration and vice versa. In addition, protective factors from a youth developmental perspective include emotional support, trust, parental control, and restrictive mediation (setting limits to, or monitoring, online activities) and evaluative mediation (open discussion about the dangers of the Internet), the lack thereof potentially creating a permissive context for cyberbullying [13]. Secondary-school adolescents join in with cyberbullying because they have more access to the Internet, have a permissive attitude towards it, and prioritise popularity over social acceptance [11,32], which concurs with the developmental theory of antisocial behaviour that adolescents exhibit more antisocial behaviour as they grow. Therefore, there is a need for a mechanism to guide cyberbullying intervention entry points in adolescents’ developmental stages. The implementation of parental control, evaluative and restrictive mediation, and trust-based and age-group-specific interventions could help address cyberbullying behaviour in schools and help to safely identify perpetrators.

### 2.10. Conceptual Framework

Studies on cyberbullying interventions are relatively new but have been increasing for the past decade, and Africa is lagging behind [22,44]. The theories that are discussed in the literature review in this study were integrated to conceptualize the problem solution. Table 1 presents a summary of theoretical characteristics that inform the conceptual framework for “the efficacy of law enforcement in addressing cyberbullying in schools”, as shown in Figure 1.

From a developmental perspective, the lack of elements such as emotional support, trust, parental control, or restrictive and evaluative mediation creates a context for cyberbullying [13]. The TPB suggests the implementation of these elements, for example, reporting should be established on efficacy beliefs [36]. Therefore, the dependent construct of this framework, “the efficacy of law enforcement in addressing cyberbullying in schools”, seeks to establish these elements. There are six independent constructs developed to empower law enforcement (police) to effectively fight against cyberbullying. The “provide safe reporting platform” and “identify cyberbullies” constructs stem from the fear of reporting cyberbullying for fear of retribution, confiscation of device, or being prevented from accessing the Internet [2,8,13,21]. Additionally, the “provide safe reporting platform” construct will facilitate restrictive mediation as a developmental element for cyberbullying interventions [13]. The anonymity provided by the technology makes it hard to identify cyberbullying perpetrators [13,17]. Therefore, the “identify cyberbullies” construct will help to identify perpetrators using peer nominations and SNA and allow targeting of specific age groups that are influential. In turn, the police can take responsibility for controlling the cyberbullying behaviour in schools. From a developmental perspective, trust is a significant element in addressing cyberbullying [13]. Therefore, the “instill trust of authorities” construct seeks to empower adolescents to safely report cyberbullying incidents and increase trust in the police. 

The developmental system and social-ecological system theories predict that creating a positive environment fosters positive behaviour [13], while the theory of planned behaviour considers social norms and attitude as the most influential elements in cyberbullying interventions [14,48]. From a social-ecological system theory perspective, the lack of behaviour-specific policies in schools permits cyberbullying [13]. On the other hand, learners’ behaviour improves when they perceive that their online activities are monitored [43]. Therefore, the purpose of the proposed framework is to enable a safe reporting platform for age-specific groups of learners and serve as surveillance awareness (restrictive mediation) to discourage adolescents from participating in cyberbullying.

Cyberbullying is a global issue, but victims still lack the cognitive and emotional tools to address cyberbullying, which necessitates timely support and intervention to curb the spread of this behaviour [21]. Therefore, “assessing cyberbullying impact” on individuals is essential to inform suitable intervention. Once the impact of cyberbullying has been established, the involved parties can be brought together to resolve their behaviour using restorative justice. The discovery of cyberbullying impact can assist with the implementation of a suitable intervention (or interventions). Convincing numerical evidence in terms of peer perceptions of cyberbullying behaviour can be presented to perpetrators using social network analysis results [34]. Then, restorative justice can be used to repair relational and social or individual harm resulting from offensive actions [53], as a suitable intervention. From a developmental perspective, restrictive and evaluative mediation is essential in addressing cyberbullying [13]. In line with the developmental systems theory [41], the creation of a positive environment by “resolving cyberbullying incidents” and “raising cyberbullying awareness” may persuade adolescents to refrain from cyberbullying behaviour, show the adverse effects of cyberbullying, and empower adolescents to report incidents.

## 3. Materials and Methods

Our researcher adopted the pragmatism philosophy as the ontological stance for a practical solution to address cyberbullying in schools [54,55]. This study required a methodology that focuses on the investigation of a solution and the understanding of its context. Therefore, the design science research methodology (DSRM) was adopted to design and investigate the artefact in context [45]. The artefact is considered the object of the study, whereas design and investigation are two major contextual activities. According to [45], design science research problems include design problems and knowledge questions. DSRM involves iterative phases of problem identification, suggestion, design, development, evaluation, and communication [56]. 

The first sections of the paper identified the problem, and a conceptual model for combating cyberbullying was presented and used to guide the research. The second section presents methods, data collection, and analysis. The first phase of data collection involved exploratory focus group discussions with key stakeholders to validate the concepts of the proposed framework. The aim of the focus group discussions was to understand participants’ firsthand and in-depth experiences regarding adolescents’ cyberbullying behaviour in schools [57,58], and discover challenges faced by the police in the fight against cyberbullying in schools to validate the proposed conceptual framework.

The second phase involved using a mobile app called mobile bully-victim response system (M-BRS) as a data-collection tool. It was necessary to develop a mobile app to test the conceptual framework, and because learners in rural schools are already using mobile phones [12], and mobile phones are the preferred Internet-access method in South African rural areas [10]. The M-BRS features include cyberbullies identification using peer and self-nominations and effects assessment among learners. 

### 3.1. Sampling Technique

Not much cyberbullying research has been conducted in the Free State province of South Africa, and ignoring such rural schools may pose high risks in curbing violence and cyberbullying [12,59]. Therefore, the purposeful sampling technique was followed in the selection of the participants in this study [5,58,60,61]. This study’s participants included eight police officers who are responsible for social crime prevention in schools. The selection of police participants in this study was based on their relevance to the research objectives and experience in their domain [56,62], to investigate their role against cyberbullying in schools. The police helped the researchers to identify schools that often had violence complaints, and teachers from these schools were involved to supplement views, due to the limited number of designated social crime prevention police in schools. Nine teachers from two of the identified schools participated in the study. Furthermore, adolescents have more Internet access than younger learners and Internet access through mobile phones is a preferred method in rural areas of South Africa [10,11,12,21], which may translate to more cyberbullying exposure. Therefore, it is important to understand cyberbullying in specific age groups of adolescents’ development [41]. Particularly, mid- and older adolescents (from grade 8) are concerned with popularity among peers and resistant to guidance from teachers than younger learners [36]. The teachers who participated in this study volunteered their classes’ learners to participate in the study.

### 3.2. Data Collection

This section presents qualitative and quantitative data-collection methods in this study. 

#### 3.2.1. Focus Group Discussion

Qualitative data collection included five focus group discussions with two groups of the police that are stationed in two separate areas, and two groups of teachers from two schools in the eastern region of the Free State province in South Africa. Identification (ID) abbreviations were assigned to the participants to enable anonymous reporting when the collected data were transcribed. The police from one area were assigned HP1 up to HPn, and the police from another area were assigned PP1 up to PPn abbreviations. Similarly, the teachers from one school were assigned IIT1 up to IITn, and teachers from another school were assigned LTT1 up to LTTn abbreviations. Table 2 presents the profile of the participants. The profile of the police officers included three females and five males, and their ages were 20–35 years (2), 36–50 years (5), and 50+ (1) years. There was one captain, three warrant officers, and four sergeants. Their experience in years of service as police ranged between 9 and 35 years. Two of the police felt that they had adequate knowledge of cyberbullying, five felt they had minimal knowledge, and only one had no knowledge of cyberbullying. Two police officers reported that they frequently dealt with cyberbullying in schools, whereas the other six rarely dealt with cyberbullying in schools. One police officer did not use social media at all, one used it rarely, and six used it frequently.

The teachers’ profile included four females and five males, and their ages were 20–35 years (6) and 36–50 years (3). Two of the teachers rarely used social media, while seven other teachers used it frequently. One of the teachers from both schools was a member of a school-based support team (SBST) that is also responsible for determining learners’ needs for support. The teachers’ experience of service ranged between 6 and 20 years. Two teachers felt that they had adequate knowledge of cyberbullying, five felt they had minimal knowledge, and two had no knowledge of cyberbullying. Three teachers frequently dealt with cyberbullying in their schools, whereas five of the other teachers rarely dealt with it, and only one teacher never dealt with it.

Due to participants’ limited knowledge of cyberbullying, our researcher used semi-structured interviews using discussion probes [63]. The following seven discussion probes were formulated based on the present study’s conceptual model constructs:Have you had a report about cyberbullying brought to you as police or teachers?If you were to verify a cyberbullying complaint, how would you be handling that?How are cyberbullying complains resolved among learners?Is there education or awareness of cyberbullying behaviour in schools or your school?Do you think that the learners trust the police or teachers to resolve their cyberbullying problems?How are emotional distressed learners helped?What can be done to encourage learners to come forward and report cyberbully behaviour?

#### 3.2.2. The Use of the M-BRS with Learners

Quantitative data were collected through the M-BRS with grade-10 and grade-12 learners for identifying the role of cyberbullying and assessing its effects. Table 3 presents participants’ demographics for grade-10 and grade-12 learners. The grade-10 group, which consisted of 52 learners who owned smartphones and obtained parents’ approval by returning signed consent forms, participated in the diagnoses of cyberbullies using the M-BRS. Their average age was 16.69 years, and 67.3% of the learners were female and 32.7% were male. Similarly, the grade 12 group consisted of 27 learners who owned smartphones and returned signed parents’ consent forms. Their average age was 19.30 years, the number of male participants was 12 (44%), and the number of female participants was 15 (56%).

To identify roles, the M-BRS presented a cyberbullying definition and a class list of learners’ names and asked learners to nominate peers that they bullied (self-nomination) or those that bullied them (peer nomination). Cyberbullying roles were identified by interpreting nominations as follows:A learner was identified as a victim if that learner nominated one or more peers as his/her bullies.A learner was identified as a bully if that learner nominated himself/herself or was nominated by one or more peers as a bully.A learner was identified as a bully-victim in a similar way as a bully but also nominated one or more peers as bullies or victims.An uninvolved learner was identified if that learner did not nominate himself/herself or was not nominated by peers.

Cyberbullying effects were also measured by asking learners to rate its impact, frequency, and content obscenity using the M-BRS. The effects assessment had three questions: (1) Please rate how obscene the bullying content was (Content obscenity); (2) Please rate the impact of the bullying incident on yourself (Impact); (3) Please rate how frequent the bullying was that happened to you (Frequency). Each question had three predefined answer-options with corresponding numerical scales: “Moderate (1)”, “Major (2)”, or “Severe (3)”. The M-BRS calculated the summary score of the selected scales to determine the level of effects (Moderate, Major, and Severe) as discussed in Section 2.5. Then, the M-BRS integrated peer and self-nominations to form network data and learners’ effects assessment to produce reports as digital trace data, and therefore acted as a data-collection tool [64,65]. The M-BRS report included the identified cyberbullying roles (bully, bully-victim, victim, and uninvolved) based on peer and self-nominations, and the degree of cyberbullying effects based on the effects assessment discussed above. It also calculated popularity as PageRank scores for each learner based on peer and self-nominations.

### 3.3. Ethical Issues

Permission to involve the police in the study was obtained from the Commissioner of the South African Police. The Department of Basic Education also gave permission to conduct the study in schools of the selected region of the Free State province in South Africa. Signed consent forms for participation and permission to record discussions were obtained from the police and teachers in this study. Similarly, learners were asked to sign and return consent forms that were also signed by their parents or guardians. As discussed in the above subsection, the learners who owned smartphones and returned signed consent forms included 79 grade-10 and grade-12 learners from one school.

### 3.4. Data Analysis

The qualitative data from focus groups were transcribed and analyzed using thematic analysis. NVivo 12 software (QSR International, Melbourne, Australia) was used to analyze data and create themes that emerged from focus-group discussions. The coding approach used in this study is deductive, which is informed by a provisional list of codes from the conceptual framework [66], and data excerpts were split into small codable moments [67]. Table 4 presents a master coding frame of four themes and six subthemes. 

The analysis of the quantitative data collected through the M-BRS’ use involved SNA and descriptive statistics. The mobile app used SNA on peer and self-nominations, and integrated nomination results with assessment data to produce reports. Data were further analyzed using Statistical Package for the Social Sciences (SPSS), Version 27 (IBM Corp, Armonk, NY, USA). The last sections of this paper discuss the findings and conclusions of this study.

## 4. Results

This study collected both qualitative and quantitative data for analysis. Section 4.1 presents the analysis of qualitative findings, and Section 4.2 presents quantitative findings.

### 4.1. Analysis of Qualitative Findings

The researchers used exploratory focus-group discussion to verify the requirements of the envisioned solution (conceptual framework) with eight police officers and nine teachers. 

#### 4.1.1. Developmental Factors

The police commented that as learners advance to secondary-school grades, their trust tends to doubt the police’s involvement in resolving the cyberbullying problem. Trust is seen as the element of emotional support, which lowers the likelihood for cyberbullying perpetration [13]. The other police commented that one of the issues affecting trust was the lack of sensitivity (emotional support) conveyed by police to learners. This could be due to the lack of clarity about the role of the police in addressing cyberbullying in schools. Therefore, there is a need to re-establish learners’ trust of authorities as capable of addressing cyberbullying cases [1], which can be achieved through the proposed conceptual framework in the current study.

Online malicious acts against teachers by their school learners (cross-age) range from insults to inappropriate picture photoshopping and impersonations. This could be due to distancing and lack of empathy [1]. Teachers reported being insulted on Facebook and feeling helpless when they could not identify culprits using interviews. The teacher commented that this instance was not reported to the police, because perpetrators could not be identified. Furthermore, cross-age cyberbullying was discovered, where learners posted inappropriate messages about their school peers’ parents on Facebook. One teacher commented that learners also reported that their parents were used as objects of cyberbullying, but she advised the reporters that nothing could be done about the report, since there was no evidence that those insults were actually posted by those suspected. Clearly, in such instances learners may feel that teachers are protecting perpetrators when they do not take prompt action about reported cyberbullying incidents, and therefore learners target teachers in attempts to avenge themselves. School violence, including bullying, escalates to retaliation against those expected to protect or take steps against perpetrators [4]. The literature states that other reasons school authorities choose not to report cyberbullying incidents include protection of their professions, avoiding stigmatisation of their schools as cyberbullying arenas, and avoiding bringing criminal charges against learners. These observations necessitate the need for a safe reporting platform, such as the proposed conceptual framework in the current study.

#### 4.1.2. Reporting Cyberbullying

The police’s experiences of cyberbullying reports varied. The police from one station stated that they do not receive reports of cyberbullying. They indicated that learners are afraid to report incidents, but they are aware of cyberbullying behaviour, whereas the police from another station indicated that they do receive reports of cyberbullying, especially online malicious acts against teachers, and their response is generally raising awareness about this behaviour using scare tactics. Teachers indicated that they learn about cyberbullying incidents by chance. When they try to address physical fights among learners, they discover that that the cause of the fight is cyberbullying. Clearly, there may be many online malicious acts against teachers and learners’ cyberbullying that remain unreported, due to the lack of a cyberbullying definition [1], fear factors for learners, and teachers’ attempts to protect their profession, stigmatisation of their schools, and trying to avoid bringing criminal charges against school learners. Therefore, learners may assume impunity and continue their cyberbullying behaviour [22], while the learners who could disclose cyberbullies fear retribution.

#### 4.1.3. Cyberbullying Role Identification

Seemingly, both the police and teachers lack adequate cyberbully-identification mechanisms. The mechanisms used to identify culprits include interrogation by the police, and interviews by teachers. The police related a story where they were investigating an online malicious incidents perpetrated by learners against teachers. Fake Facebook accounts were used to insult teachers. One suspect was identified because he used a valid Facebook account. However, he was threatened with violence if he disclosed his accomplices to the police. The suspect claimed that he was lured to participate in the act using a valid identity on Facebook, while his accomplices used fake identities. The literature reports fear of retribution and perpetrator anonymity as major factors for lack of reporting cyberbullying among learners. The police also stated that they rely on social workers’ counseling because learners who engage in cyberbullying often do not see any wrongdoing in their behaviour. Furthermore, the police appreciated the suggestion of using peer nominations as a mechanism to safely identify cyberbullying perpetrators and victims among learners, and added that learners’ parents must first provide consent. This observation reiterates the need to identify cyberbullies through peer nominations and presenting nomination numbers as evidence of peers’ perceptions about perpetrators’ behaviour. 

#### 4.1.4. Awareness

The results showed that the police use scare tactics including risk of incarceration and criminal records as awareness methods. Teachers use the school policy that prohibits learners from bringing mobile phones to school to raise awareness of the dangers of cyberbullying, but learners do not adhere to this policy. The police reported that they requested the school principal to allow one of the researchers in the current study to address learners as an initiative to raise cyberbullying awareness. Other police suggested that the researchers in the current study should go with them to other schools to raise awareness. These observations imply that the police and teachers lack adequate awareness initiatives and skills in relation to cyberbullying.

#### 4.1.5. Resolving Incidents

The participants’ approach to resolving cyberbullying incidents include talking to learners and warning them about damages of cyberbullying to others, applying restorative justice. Restorative justice may be suitable for resolving aggression between learners, as it focuses on repairing relational and social or individual harm resulting from offensive actions [53]. However, when using restorative justice only, learners may feel their cyberbullies do not receive well-deserved punishment for their actions, and may therefore avenge themselves by resorting to violence. Hence, in addition to the use of restorative justice, addressing cyberbullying through punitive measures may increase reporting for victims [51], which may help to avoid retaliation. Moreover, this observation indicates that learners do not understand interventions or rules used to resolve cyberbullying incidents and their purpose [13,20]. Clearly, both the police officers and teachers would benefit from training to address cyberbullying. The authors of [21] suggest that preventive programmes should be built with a shared teachers’ commitment for effective implementation that is often limited by time for outsiders. 

#### 4.1.6. Effects Assessment

The important element of interventions is gauging the impact of cyberbullying to determine suitable entry points of intervention [68]. The police and teachers indicated that they rely on social workers to assess the impact and counsel victims of cyberbullying. However, this method is not effective due to time and capacity constraints in the Department of Social Development in rural areas such as the eastern region of the Free State province in South Africa [2,25]. This observation necessitates a prompt method to determine the impact of cyberbullying on victims and recommend a suitable intervention.

The police and teachers’ knowledge of cyberbullying could be enhanced with mechanisms to identify cyberbullies and measure the impact of cyberbullying to inform suitable interventions. This enhancement is particularly essential because the police and teachers’ demographics in the present study indicated that their knowledge of cyberbullying behaviour was minimal. Perpetrators often fear disclosing their accomplices, and investigation methods used to identify suspects risk informants’ victimisation as “snitches” who “sell out” their accomplices. These methods have no safety assurance for informants who wish to report cyberbullies or their accomplices. Therefore, the challenge of identifying cyberbullies was highlighted in all participants’ discussions. These observations support the need for the proposed conceptual framework for the identification and determination of the impact of cyberbullies as a restrictive mediation among school learners.

### 4.2. Analysis of Quantitative Findings

Descriptive statistics were calculated by the mobile app [69], as well as SAN to provide mathematical analysis of the reports [70]. This subsection presents M-BRS’ reports including centrality (PageRank), effects, cyberbullying role (bully, bully-victim, victim, and uninvolved) for the grade-10 and grade-12 learners. Two reports were produced by the M-BRS, one for grade-10 learners and another one for grade-12 learners.

#### 4.2.1. Cyberbullying Roles

The M-BRS was used to allow learners to identify their cyberbullies and victims through peer and self-nominations as presented in Section 3.2.2. Doing so helped to identify the cyberbullying roles of the learners who participated in this study. Table 5 presents a summary of the identified cyberbullying roles among grade-10 and grade-12 learners. Of the 52 grade-10 learners, 17 (33%) were identified as bully-victims, 10 (19%) as victims, 11 (21%) as bullies, and 14 (27%) as uninvolved. Of the 27 grade-12 learners, 14 (52%) bully-victims were identified, 5 (19%) bullies, 3 (11%) victims, and 5 (18%) uninvolved learners. The authors of [13] note that the lack of clear rules, parental control, and restrictive and evaluative mediations permit cyberbullying behaviour in schools. This observation shows that both mid- and older adolescents are more likely to report cyberbullying incidents using a safe reporting platform without fear of retribution; such a platform could serve as a restrictive mediation in schools. Furthermore, the police can use these results as convincing evidence to perpetrators of their unacceptable behaviour. The chi-square test suggests no statistically significant difference in the cyberbullying role between grade-10 and grade-12 groups, X^2^ (3) = 2.95, *p* > 0.05.

#### 4.2.2. Cyberbullying Effects

Table 6 presents the report of cyberbullying effects for both grade-10 and grade-12 learners. Among grade-10 learners, 9 of the 17 mobile bully-victims experienced moderate effects and 8 experienced severe affects. Similarly, 5 of the 10 victims reported moderate effects, and the others reported severe effects. None of the bully-victims and victims reported major effects. Among grade-12 learners, 8 of the 14 bully-victims reported moderate effects, and the other 6 reported major effects. All three victims experienced major effects. Grade-12 learners indicated moderate and major effects only, whereas grade-10 bully-victims and victims indicated moderate and severe effects only. We could not do a chi-square test of the cyberbullying effects differences between grade-10 and grade-12 groups, since in some cases no effects were reported, as indicated in Table 6.

#### 4.2.3. Cyberbullying Popularity

A summary of learners’ popularity among grade-10 and grade-12 learners is presented in Table 7. PageRank scores represent cyberbullying popularity as discussed in Section 2.3, and the M-BRS calculated these scores from peer nominations. These scores signal prioritisation for interventions based on cyberbullying popularity. A two-way ANOVA was performed to compare whether gender and grade or their interaction were statistically significant in their effect on learners’ PageRank scores. There was a statistically significant difference between grade 10 and grade 12 on learners’ PageRank scores (F(1,75) = 6.01, *p* < 0.02). Grade-12 learners had higher PageRank scores, with a mean of 0.86 and standard deviation (SD) of 0.43, than grade-10 learners (mean = 0.56, SD = 0.55), which is consistent with suggestions that the concern for popularity among mid- and older adolescents is high, but low among younger adolescents [32,36]. Therefore, grade-12 learners could be the first target group for specific age-group interventions. However, there was no statistically significant difference in gender on learners’ PageRank scores (F(1,75) = 0.72, *p* > 0.05) with males (mean = 0.74, SD = 0.49) and females (mean = 0.62, SD = 0.56). Additionally, there was no statistically significant difference between gender and grade (F(1,75) = 0.74, *p* > 0.05).

## 5. Discussion

From a developmental perspective, the results of the qualitative data showed that older adolescents do not trust the police to handle cyberbullying incidents, and further showed that this is due to the police’s lack of emotional support for adolescents. Trust is an element of emotional support in a developmental context that is necessary in addressing cyberbullying [13]. The results of the quantitative data show that adolescents are more likely to report cyberbullying incidents using a safe reporting platform when they understand the aim of using such platforms. This observation reaffirms suggestions that learners need to remain anonymous to report incidents [71], and therefore, as suggested in the conceptual framework in this study, using safe reporting platforms could help to instill learners’ trust in the police.

As also noted in the research [1], the qualitative results of this study showed that the learners’ violence against teachers also manifests through online malicious acts. The qualitative results indicate that teachers are often targeted by learners online (cross-age malicious acts) because they do not take prompt actions against perpetrators. However, the police and teachers lack mechanisms to identify cyberbullying culprits among learners. Therefore, training on mechanisms to address the cyberbullying behaviour in schools could be beneficial for the police and teachers. Furthermore, the quantitative data showed higher degrees of cyberbullying popularity for high-school learners in grade 12 than those in grade 10. This is because adolescents try to entertain themselves online without realising the significance of their actions to others [72]. As also noted by [47], age and class justification (shared belief) of cyberbullying are developmental effects that significantly influence adolescents. Therefore, these observations support the developmental systems theory in that high-school learners in higher grades may also target their teachers as means to gain popularity, if they believe it is justified, whereas high-school learners in lower grades may be less inclined to target their teachers online. The results reaffirm the view that adolescents perpetrate bullying through social networking sites as they gain greater Internet access [11,13], which validates the proposed conceptual framework to target specific age groups for combating cyberbullying in schools.

The quantitative data results showed, from a developmental perspective, that the degree of cyberbullying effects was higher for learners in lower grades than for learners in higher grades in schools. This observation suggests that cyberbullying effects may not be similar for both mid- and older adolescents. Therefore, from developmental theory perspectives in conjunction with socio-ecological theory and the theory of planned behaviour, subjective norms (such as justifying negative behaviour) and lack of clear rules influenced mid- and older adolescents’ involvement in cyberbullying [33,47,48]. From a developmental perspective, affection goals (being loved and intimate) affect status goals and peer cyberbullying for older adolescents, and therefore they attempt to gain peers’ affection or attention through delinquency, but younger adolescents still depend on parental love [33]. That is, young adolescents are not concerned with being socially accepted or popular for acceptable behaviour among peers. These observations call for age-group-specific interventions to ensure efficacy.

Emerging patterns in the developing countries indicate where parents and teachers lack skills and support regarding Internet use and awareness, learners engage in more risky online behaviour, such as contacting, and sharing pictures and personal information with strangers [21,73]. Furthermore, we are cognizant of the fact that violence such as school shootings also springs from a lack of interventions for bullying from those expected to ensure safety [4]. Considering these observations, researchers recommend the use of a safe reporting system as an intervention. Additionally, the police, together with teachers, should be trained and capacitated by focusing on the comprehensive specifications and breakdown of cyberbullying. Therefore, prevention programmes should not just sensitise parents, teachers, and police about cyberbullying but “better prepare them for prevention and interventions” [21] (p. 937).

## 6. Limitations

This study presented the developmental issues contributing to cyberbullying and the police response to this violence in rural schools. The sample size presents some limitations to this study. There were a limited number of police officers responsible for social crime prevention in schools; therefore, future studies should aim for larger numbers of participants. The limited number of participants also limits the generalisation of the findings. In addition, we could not perform a chi-square test for the cyberbullying effects differences between grade-10 and grade-12 learners because some cases of cyberbullying effects were not reported. Therefore, caution needs to be exercised when interpreting some of the results.

## 7. Conclusions

Following the pragmatism philosophy as an ontological stance to investigate a practical solution in this study allowed researchers to adopt a design science research approach to create an innovative solution to address cyberbullying in schools. This approach allowed researchers to develop a conceptual framework from the literature review and test the conceptual framework using suitable methods. These methods include the use of exploratory focus-group discussions and the mobile response system (M-BRS) as data-collection tools, and the use of both qualitative and quantitative analyses to clarify results.

The conceptual framework was developed by integrating theoretical frameworks, which include the theory of planned behaviour, DST, and socio-ecological system theory. This conceptual framework provides a more comprehensive analysis of the cyberbullying challenge in rural-area schools. The proposed conceptual framework will be useful to the police in implementing and monitoring age-group-specific interventions. The police can use technologies such as the M-BRS to safely identify cyberbullies and assess more effectively the effects of cyberbullying among learners in school.

The researchers found that mid- and older adolescents do not trust the police to handle cyberbullying incidents because the police do not provide adequate emotional support. It was also found that teachers are often targeted by learners online (cross-age online malicious acts) because they do not act promptly when cyberbullying incidents are reported to them. Additionally, quantitative results showed that cyberbullying popularity was higher for learners in higher grades than for those in lower grades. This is because older adolescents are more concerned about gaining popularity than being socially accepted [36]. These observations confirm age and cyberbullying popularity as developmental issues influencing cyberbullying in schools. Furthermore, these findings are consistent with the DST, which posits that the behaviour of learners is a result of interactions with the environment without being taught how to behave (self-organisation process) [52]. Therefore, the lack of a supportive context, such as emotional support or restrictive mediation, creates a permissive context for cyberbullying [13].

The finding of this study showed that cyberbullying popularity is a strong influencing factor among older adolescent groups. Therefore, researchers recommend the use of a safe reporting system as a cyberbullying intervention among adolescents’ specific age groups. Additionally, the police, together with teachers, should be trained and capacitated by focusing on the comprehensive specifications and breakdown of cyberbullying. In the context of South African rural-area schools, the police also need training to practise emotional support. This will better prepare the police in relation to age-group-specific interventions and prevention programmes [21], and will enable them to leverage mobile apps to monitor cyberbullying in rural schools.

## Figures and Tables

**Figure 1 ijerph-18-13421-f001:**
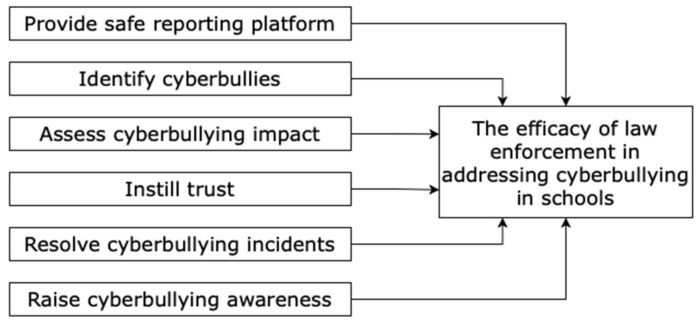
Conceptual framework for effective cyberbullying combat.

**Table 1 ijerph-18-13421-t001:** Guiding theory characteristics.

Theory	Characteristic	*Construct*	References
TPB	Drive school climate change, social norms, attitude, and efficacy	Identify cyberbullies, assess impact and reports Resolving cyberbullying incidents	[11,14,36,48]
SES	Cyberbullying permissible context	Raising cyberbullying awareness	[13,49,50]
DST	Environmental and personal traits interactions influence envelopment	Provide safe reporting platformInstill trust of authorities	[11,13,41,52]

**Table 2 ijerph-18-13421-t002:** Police and teachers’ profile.

ID	Gender	Age Group	Rank or Occupation	Experience (years)	Knowledge Level of Cyberbullying	Frequency of Dealing with Cyberbullying	Social-Media Usage
HP1	Male	36–50	Warrant officer	31	Adequate	Rarely	Frequently
HP2	Male	36–50	Sergeant	16	Minimal	Rarely	Never
PP1	Female	36–50	Captain	35	Adequate	Frequently	Rarely
PP2	Male	36–50	Warrant officer	29	Minimal	Frequently	Frequently
PP3	Male	36–50	Warrant officer	24	Minimal	Rarely	Frequently
PP4	Male	50+	Sergeant	30	None	Rarely	Rarely
PP5	Female	20–35	Sergeant	9	Minimal	Rarely	Frequently
PP6	Female	20–35	Sergeant	11	Minimal	Rarely	Frequently
LTT1	Male	20–35	Teacher	7	Minimal	Rarely	Frequently
LTT2	Male	20–35	Teacher	9	Minimal	Rarely	Frequently
LTT3	Female	20–35	Teacher (SBST)	6	Adequate	Frequently	Frequently
LTT4	Male	36–50	Teacher	20	None	Never	Rarely
IIT1	Female	20–35	Teacher	6	Minimal	Rarely	Frequently
IIT2	Female	20–35	Teacher (SBST)	8	Adequate	Frequently	Frequently
IIT3	Male	36–50	Teacher	23	Minimal	Rarely	Rarely
IIT4	Female	36–50	Teacher	20	None	Rarely	Frequently
IIT5	Male	20–35	Teacher	7	Minimal	Frequently	Frequently

**Table 3 ijerph-18-13421-t003:** Learners’ demographics.

Grades	Youngest	Oldest	Males	Females	Average Age	Standard Deviation
Grade 10 (N = 52)	15	19	17	35	16.69	1.00
Grade 12 (N = 27)	17	23	12	15	19.30	1.48

**Table 4 ijerph-18-13421-t004:** Sample responses from the police and teachers.

Themes	Comments
**Developmental**	
Trust	“*Learners below grade 6 are not such a big problem but once they start from grade 6, 7, 8 they start to have doubts about adults*” (HP1). “*Other (police) departments when they go and address the learners, they are not concentrating on their feelings they just tell them what they want*” (HP2).
Cross-age	“*We get reports that there are some learners who are bullying them through Facebook. They are talking bad things about their parents and bad things about their lives*” (IIT1). “*Mam, they wrote about my mom (on Facebook), I’m so angry, and if I find them, those two girls, I’m going to fight them because they wrote about my parent*” (IIT2). “*Let us take for instance this one (investigation) of today, we come from the class now. That guy (learner) is afraid of those people who have created the account (Facebook account) on his name*” (PP1). “*The teachers were talking to learners, where they actually did not get to a solution or to even say who did it*” (LTT3).
Cross-platform	“*They (perpetrators) are bullies from the class itself, it’s a threat to him (suspect or victim). He (suspect or victim) knows that when he reports. When he comes back those people are going to do whatever they want to him. It’s a fear*” (PP1). “*I think if we can cut (stop) gangsters at schools. Some of them [learners] are being bullied, and afterwards there are gangsters maybe that are controlling other learners, maybe they are afraid to report this cyberbullying*” (PP4). “*You hear somebody else saying these (learners) are fighting because this one wrote such and such (on Facebook) about the other*” (LTT3).
**Inadequate awareness**	
	“*It was raised by some of them (learners) that there is mobile bullying in schools, they did not know what the name is for it, but they described bullying by cell phones and WhatsApp, and I said, it is called mobile bullying*” (HP1).
	“*Most of the learners of the school were not aware of these cybercrimes. They are using the computer and cell phones”* (PP2). ” … *we informed the principal that we will bring someone to address all learners*” (PP2). “*We need to go to more schools and present this cyberbullying, so they can know more about it (cyberbullying). In that way they will be able to come to the police station and report it if it is happening to them. So, they need to get more knowledge about this cyberbullying*” (PP5). “*there is a school policy, we do not allow phones at school, but you know the learners*” (IIT3).
**Resolving incidents**	
Scare tactics	“*As for now we just use (scare) tactics to say this and that will be done on the perpetrators [to discourage their behaviour]. However, it is going to be a difficult thing at some stage because we don’t have full equipment for that. We just tackle it with fear*” (PP1).
Restorative justice	“*I decided to talk to them privately and say to them you know what you do not have full evidence*” (IIT2). “*In actual fact, they are minors we cannot do anything harsh on them*” (PP1). “(We) *rely on witnesses and maybe on injuries if it was physical bullying, but if it was cyberbullying, they can maybe show you the message on the phone so that you can read it for yourself*” (HP1). “*So, we managed to get that culprit on Tuesday, and try to show him that this is a not okay*” (PP2). “*We will call both pupils and address them and show the other party the results of bullying others. For me I think that one has helped a lot in our area*” (HP2). “*I will call the victim and the perpetrator, sit them down and show them the consequences of cyberbullying*” (HP2). “*When the issue cannot be resolved parents are called in*” (HP1).
**Identification**	
Peer nominations	“*I like the idea and agree with you 100%, because it is a closed system, no one will know except for us (police), when we assess to say, okay ten of the pupils identified (nominated) one person, meaning that person is a suspect. We can say this person is the culprit*” (PP1). “… *but when parents come, they (parents) would like to know who did this to my child. Since they are minors*” (PP2). “*Sometimes for these learners it is difficult to accept that they are doing wrong, so that is why we involve the Social Workers*” (HP2).
Effects assessments	“*Even when it comes to referrals themselves, they can take forever, but it’s only so much that they can do*” (LTT3) “*Depending on how great the harm is. They (teachers) refer the issue to the district, and when learners are referred to the district, they are assisted whichever way they need*” (LTT3). “*Even when it comes to referrals themselves (they) can take forever*” (LTT3).
**Reporting**	
Unplanned	“*We just heard the rumors that these (learners) did that*” (IIT1). “*You hear somebody else saying these (learners) are fighting because this one wrote such and such* (*on Facebook*) *about the other*” (LTT3).
Fear	“*It [encouraging learners to report incidents] might be the problem, because some of the learners…, so they are afraid to come forward to report that crime*” (PP2).

**Table 5 ijerph-18-13421-t005:** Cyberbullying roles.

Grade	Bullies	Bully-Victims	Victims	Uninvolved
Grade 10 (N = 52)	11 (21%)	17 (33%)	10 (19%)	14 (27%)
Grade 12 (N = 27)	5 (19%)	14 (52%)	3 (11%)	5 (18%)

**Table 6 ijerph-18-13421-t006:** Cyberbullying effects.

Grade	Role	Moderate	Major	Severe
Grade 10	Bully-victims	9 (53%)	0 (0%)	8 (47%)
	Victims	5 (50%)	0 (0%)	5 (50%)
Grade 12	Bully-victims	8 (57%)	6 (43%)	0 (0%)
	Victims	0 (0%)	3 (100%)	0 (0%)

**Table 7 ijerph-18-13421-t007:** PageRank (Popularity) summary.

Gender	Grade	Mean	SD
Female	10	0.62	0.55
Male	12	0.74	0.49
Total	10	0.56	0.55
	12	0.86	0.43

## Data Availability

Data are available on request due to ethical restrictions. The data presented in this study are available on request from the corresponding author.

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
