# Peer review of "The Response of Social Crime Prevention Police to Cyberbullying Perpetrated by Youth in Rural Areas of South Africa"

_ijerph, 2021, doi:10.3390/ijerph182413421_

Round 1

Reviewer 1 Report

Good job! Very interesting and is a good contribution to understanding the phenomenon of cyberbullying. Most definitions of bullying and cyberbullying usually refer to aggressions carried out "between equals", so I suggest reviewing the name given to aggressions that are from students to teachers. Perhaps pose it as another modality within school violence. In addition, in the same definition that is offered of cyberbullying, “repetition in time” is indicated as a characteristic and this is not clear if it occurs in the attacks against teachers who are cited in your work.

Some quotes are identified that do not necessarily support the claim made. For example, in paragraph 1 on page 2: “Although there are many studies on peer bullying, developmental patterns and cross-age (learner-to-teacher) bullying are not well researched [7,9,13]” the quote refers to the prevalence of cyberbullying among peers of different ages, but does not measure student aggression against teachers.

On the other hand, it is not clear with how many and which questions the effect/impact of cyberbullying was evaluated.

Author Response

The authors thank the reviewers for taking the time to review the manuscript and their invaluable comments and suggestions. Please find the attached schedule of revisions effected in the manuscript.

Reviewer 2 Report

  1. On page 2, line 55, I have some trouble understanding the context of this first sentence on cyberbullying. Is this about general information, specific to this population, or this environment of the school? Perhaps clarifying this would be beneficial.
  2. In the sampling, it is suggested that teachers gave consent on behalf of students. Whether the 97 student participants themselves gave consent or assent or neither should be clarified. Later in the manuscript, it suggests that those in grade 12 were provided consent forms, but that is only 27 cases.
  3. What is the role of police in schools in South Africa? Are they permanently assigned or seconded from departments? Briefly clarifying earlier in the manuscript how this arrangement is structured would add make it more accessible.
  4. In the results, what testing is suggesting there is a significant difference between these groups? Could a test not be used in the discussion of the results of the app (e.g. moderate, severe) to determine whether these differences are significantly different by incorporating the sample size? The same could be done for the grade differences.

Author Response

(The authors gave the same response as above.)
